# The Crosstalk between Src and Hippo/YAP Signaling Pathways in Non-Small Cell Lung Cancer (NSCLC)

**DOI:** 10.3390/cancers12061361

**Published:** 2020-05-26

**Authors:** Ping-Chih Hsu, Cheng-Ta Yang, David M. Jablons, Liang You

**Affiliations:** 1Department of Surgery, Helen Diller Family Comprehensive Cancer Center, University of California, San Francisco, CA 94115, USA; 8902049@gmail.com (P.-C.H.); david.jablons@ucsfmedctr.org (D.M.J.); 2Division of Thoracic Medicine, Department of Internal Medicine, Chang Gung Memorial Hospital at Linkou, College of Medicine, Chang Gung University, Taoyuan 33305, Taiwan; Yang1946@cgmh.org.tw; 3Department of Respiratory Therapy, College of Medicine, Chang Gung University, Taoyuan 33302, Taiwan

**Keywords:** proto-oncogene tyrosine-protein kinase Src, yes-associated protein (YAP), Hippo pathway, non-small cell lung cancer (NSCLC), tyrosine-kinase inhibitor (TKI), YES1, dasatinib

## Abstract

The advancement of new therapies, including targeted therapies and immunotherapies, has improved the survival of non-small-cell lung cancer (NSCLC) patients in the last decade. Some NSCLC patients still do not benefit from therapies or encounter progressive disease during the course of treatment because they have intrinsic resistance, acquired resistance, or lack a targetable driver mutation. More investigations on the molecular biology of NSCLC are needed to find useful biomarkers for current therapies and to develop novel therapeutic strategies. Src is a non-receptor tyrosine kinase protein that interacts with cell surface growth factor receptors and the intracellular signaling pathway to maintain cell survival tumorigenesis in NSCLC. The Yes-associated protein (YAP) is one of the main effectors of the Hippo pathway and has been identified as a promoter of drug resistance, cancer progression, and metastasis in NSCLC. Here, we review studies that have investigated the activation of YAP as mediated by Src kinases and demonstrate that Src regulates YAP through three main mechanisms: (1) direct phosphorylation; (2) the activation of pathways repressing Hippo kinases; and (3) Hippo-independent mechanisms. Further work should focus on the efficacy of Src inhibitors in inhibiting YAP activity in NSCLC. In addition, future efforts toward developing potentially reasonable combinations of therapy targeting the Src–YAP axis using other therapies, including targeted therapies and/or immunotherapies, are warranted.

## 1. Introduction

Lung cancer is the leading cause of cancer-related deaths worldwide, and most patients are diagnosed at an advanced stage of disease [1,2]. Non-small-cell lung cancer (NSCLC) accounts for 85% of lung cancer cases, according to the classification of histopathology [3,4]. Chemotherapy is used as the standard treatment for most patients with advanced disease, but its effect is limited because of unsatisfactory efficacy and side effects [5]. In the last decade, new therapies that have been developed for advanced NSCLC treatment, including targeted therapy and immunotherapy, have shown better clinical effects in terms of the objective response rate and prolonging survival in comparison with conventional chemotherapy [6,7]. Several oncogenic drivers of NSCLC, such as the epidermal growth factor receptor (EGFR) and BRAF mutations, as well as anaplastic lymphoma kinase (ALK) and ROS-1 rearrangements, have been uncovered and studied. Drugs targeting these oncogenic drivers have been developed, and most of them have been approved by the US Food and Drug Administration (FDA) for the clinical treatment of advanced NSCLC patients harboring oncogenic-driven mutations [6,8,9]. Currently, targeted therapies, including those based on the use of EGFR-tyrosine kinase inhibitors (TKIs), BRAF inhibitors, and ALK inhibitors, have demonstrated promising efficacy, with a 60–80% response rate and 9–30 months of progression-free survival in treating advanced NSCLC with relevant driver mutations [6,8,9]. Immunotherapy using anti-PD-1/PD-L1 immune checkpoint inhibitors has been developed as a new therapy for metastatic NSCLC that has demonstrated efficacy in numerous clinical trials in which a 15–45% objective response rate was observed in addition to prolonged overall survival [7,10,11,12]. However, some NSCLC patients still do not have targetable driver mutations and do not benefit from immunotherapy [13]. In addition, acquired resistance to targeted therapies and immunotherapy occur after a period of treatment in most NSCLC patients who had an initial response to the treatment [13]. Acquired resistance to multiple treatment modalities limits the five-year survival rate of NSCLC patients (~16%) [2,3,4,5,13]. Therefore, further investigation on the molecular biology of NSCLC is needed for the development of novel therapeutic strategies and the determination of useful biomarkers for treatment selection.

Src (also known as proto-oncogene tyrosine protein kinase or c-Src) is a non-receptor tyrosine kinase protein that has been reported to promote tumorigenesis and metastatic progression in various cancers [14,15]. Src has also been identified in human NSCLC and has been reported to interact with several growth factor receptors including EGFR, human epidermal growth factor receptor 2 (HER2), hepatocyte growth factor receptor (HGFR; also known as c-MET), platelet-derived growth factor receptor (PDGFR), insulin-like growth factor receptor (IGFR), fibroblast growth factor receptor (FGFR), vascular endothelial growth factor receptor (VEGFR), and focal adhesion kinase (FAK) [14,15]. Yes-associated protein (YAP) is a key mediator of the Hippo signaling pathway and has been identified as an important oncoprotein in human NSCLC [16,17]. YAP has been reported to interact with growth factor receptors and their downstream signaling pathways to promote drug resistance, cancer progression, and metastasis in NSCLC [18,19]. 

Here, we review the crosstalk between Src family kinases and the Hippo/YAP signaling pathway and look for new therapeutic targets in human NSCLC for future study. 

## 2. Src in NSCLC

Src is a non-receptor tyrosine kinase protein that consists of the SH1 (tyrosine kinase) SH2, SH3, and SH4 (unique) domains; the SH3–SH2 connector; the SH2–kinase linker; a C-terminal tail regulatory region; and two tyrosine sites (Tyr416 and Tyr527) (Figure 1A.) [14,20]. The tyrosine site Tyr416 is located in a kinase domain (SH1), while the other tyrosine site, Tyr527, is located in the C-terminal region [14,20]. The intramolecular interactions among the domains and phosphorylation of the two tyrosine sites are crucial for the regulation of Src. Src is usually present in an inactive form when phosphorylated Tyr527 binds to the SH2 domain and the SH2–kinase linker binds to the SH3 domain. This conformation protects the catalytic pocket of Tyr 416 in the kinase domain (SH1) from inappropriate phosphorylation. The dephosphorylation of Ty527 causes a conformational change that unlocks the catalytic pocket of Tyr 416 and leads to subsequent activation of Src by the intramolecular autophosphorylation of Tyr416 [14,20,21] (Figure 1B). 

Src protein kinase is expressed in the cells of normal tissues including the brain, bone, lungs, leukocytes, and platelets. Src is involved in the transduction of signals from the cell surface receptors and plays an important role in regulating cell growth, differentiation, and survival [22,23,24]. Regarding the regulation of intracellular signaling transduction, Src has been identified as an important oncoprotein that promotes cancer progression, invasion, metastasis, and drug resistance in various cancers, including colorectal cancer, breast cancer, pancreatic cancer, gastric cancer, and lung cancer [14,15,25,26,27]. The overexpression and/or hyperactivation of Src have been reported in human cancers and lead to the upregulation of various receptors of tyrosine kinases (RTKs), including EGFR, HER2, c-MET, PDGFR, IGFR, FGFR, and VEGFR [14,15]. In addition, Src interacts with other transmembrane receptors, such as integrins/Focal adhesion kinase (FAK) and G-protein-coupled receptors (GPCRs), to upregulate their downstream signaling [28,29,30]. Regarding human NSCLC, previous studies have shown that the overexpression of Src is associated with poor prognosis and the promotion of migration, invasion, metastasis, and drug resistance [31,32,33,34]. EGFR kinase domain mutations, such as L858R and exon 19 deletion, are the most frequent oncogenic-driven mutations in human NSCLC (5–15% in Caucasians and 40–55% in East Asians), and EGFR-tyrosine kinase inhibitors (TKIs) have been developed as an effective therapy for advanced EGFR-mutated NSCLC [35,36,37]. EGFR kinase domain mutations activate downstream signaling pathways, including the mitogen-activated protein kinase (MAPK)/extracellular signal-regulated kinase (ERK), phosphatidylinositol 3-kinase (PI3K)/Akt/mTOR, and interleukin 6 (IL-6)/Janus kinase (JAK)/signal transducer, and the activator of transcription 3 (STAT3) signaling pathways [34]. Several previous studies have provided evidence that Src activates the three downstream signaling pathways (MAPK/ERK, PI3K/Akt/mTOR, and IL-6/JAK/STAT3), thus promoting resistance to EGFR-TKIs in NSCLC [38,39]. A recent study found that the Src family kinase YES1 (v-YES-1 Yamaguchi sarcoma viral oncogene homolog 1) was amplified in NSCLC patients harboring EGFR mutations or echinoderm microtubule-associated protein-like 4 (EML4)-anaplastic lymphoma kinase (ALK) fusion who had acquired resistance to EGFR or ALK inhibitors [40]. Therefore, the amplification of the Src family kinase YES1 was identified as a mechanism for acquired resistance to EGFR and ALK inhibitors. In a study by Fan et al., it was demonstrated that the forced overexpression of YES1 promotes resistance to EGFR-TKIs in the human NSCLC cell line PC9, and the inhibition of Src re-enhances the cytotoxicity of EGFR-TKIs [40]. Two previous studies also demonstrated that inhibiting Src family kinase enhances the antitumor effect of EGFR-TKIs on EGFR-mutated NSCLC cells [38,41]. Studies about Src-altered resistance to targeted therapy in NSCLC are summarized in Table 1.

G-protein-coupled receptors (GPCRs) constitute a large family of diverse receptors that play crucial roles in the function of normal tissues through the activation of intracellular signaling pathways in response to specific extracellular signals [28]. GPCRs have been found to play an important role in tumorigenesis and cancer progression and have been reported to be overexpressed in human NSCLC [42,43]. Therefore, therapeutic approaches targeting GPCRs in metastatic NSCLC have been developed, and their efficacy is currently under investigation [44,45]. Previous studies have shown that the downstream signaling transduction of GPCRs depends on the phosphorylation of Src kinase in some manner [46,47]. Another two previous studies demonstrated that the crosstalk between GPCRs and EGFR is bridged by Src kinase in human NSCLC cells. They found that some GPCRs, such as formyl peptide receptor-like 1 (FPRL1) and cholecystokinin (CCK) receptors, are present in human NSCLC cells and transactivate EGFR downstream signaling pathways, including MAPK and JAK/STAT3, to promote cell proliferation through Src kinase. In addition, both studies showed that the inhibition of Src blocks the transactivation of EGFR induced by the GPCRs [48,49]. The Rho/Rho-associated protein kinase (ROCK) pathway is a downstream signaling pathway that is regulated by Src and is involved in the promotion of cancer invasion and metastasis [50]. The co-expression of Rho and Src in human NSCLC has been reported, and therapy based on the targeting of Rho/ROCK in metastatic NSCLC has been investigated in several studies [31,51,52,53]. Onodera et al. showed that the concurrent inhibition of Src and Rho/ROCK has a synergistic effect in suppressing NSCLC cell growth [53].

The Src kinase protein plays an important role in mediating signaling transduction between transmembrane cell surface receptors and downstream intracellular pathways. The gene amplification of YES1 (one of the Src family kinases) has been found to partly impact the clinical prognosis of stage I or II NSCLC in previous studies [54]. To date, YES1 is the only SRC family kinase member that has been found to be regulated by gene amplification, and a high correlation has been demonstrated between the gene copy number and mRNA expression in human NSCLC [55]. Further, the experimental findings of a recent study conducted by Garmendia et al. suggest that molecular alterations of YES1 could be used as a prognostic biomarker and therapeutic target in human NSCLC [56]. First, they showed that NSCLC patients with high YES1 expression had significantly shorter overall survival times than those with low YES1 expression. Second, YES1 overexpression significantly increased NSCLC cell proliferation in vitro and induced metastatic spread in preclinical mouse models. Third, the knockdown of the YES1 gene decreased NSCLC cell proliferation, invasion ability, and tumor growth in vivo. In addition, the antitumor effect of the Src inhibitor dasatinib was investigated. It was found that dasatinib has significantly higher cytotoxicity towards NSCLC cells with high rather than low YES1 expression. Furthermore, dasatinib treatment was found to significantly inhibit tumor growth in a patient-derived xenograft (PDX) model with high YES1 expression. In these PDX models, dasatinib treatment did not affect tumor growth in patients with low YES1 expression. The study suggests that selected advanced NSCLC patients with high YES1 expression or genetic amplification may benefit from dasatinib treatment [56].

The interactions among transmembrane cell surface receptors, Src kinase, and downstream signaling pathways are summarized in Figure 2.

## 3. Hippo/YAP Singing Pathway in NSCLC

The Yes-associated protein (YAP) is one of the core effectors of the Salvador–Warts–Hippo (or simply Hippo) signaling pathway, and the other core effector is the transcriptional co-activator with a PDZ-binding motif (TAZ) [57]. In normal organ tissues, the Hippo/YAP pathway functions to regulate organ size growth control, stem cell function, and regeneration [16,57,58]. Deregulation of the Hippo pathway and hyperactivation of YAP are frequently found in a diverse range of cancers, and the Hippo/YAP pathway has been suggested to be involved in cancer initiation and progression [16,57]. YAP is negatively regulated by upstream components of the Hippo pathway (known collectively as the Hippo kinases), including neurofibromatosis 2 (NF2), large tumor suppressor homolog 1 (LATS1), LATS2, and mammalian sterile 20-like kinase 1 (MST1). Usually, activation of the Hippo pathway leads to tumor suppression, and Hippo kinase sequesters and degrades YAP in the cytoplasm. Conversely, when the Hippo pathway is deregulated, there is an increase in the translocation of cytoplasmic YAP into the nucleus to form complexes with transcriptional enhancer factors (TEFs; also known as TEAD). In cancer cells, the binding of YAP and TEAD in the nucleus activates the transcription of downstream genes to regulate cell proliferation, the epithelial to mesenchymal transition (EMT), metastasis, cell survival, drug resistance, and cancer stem cell characteristics [16,57]. The loss of the Hippo pathway through mutation and/or the downregulation of core Hippo components has been found in various cancers and results in elevated levels of nuclear-localized YAP. YAP plays a crucial role in oncogenic transformation and has been reported to promote cancer development in various human cancers, including lung, malignant pleural mesothelioma, breast, liver, melanoma, colon, and urogenital cancers [59,60,61,62,63,64]. Low expression of the Hippo kinase LATS1 in addition to YAP overexpression have been identified in human NSCLC and are associated with poor prognosis [65,66]. Many previous studies have shown that YAP plays an important role in promoting the epithelial–mesenchymal transition (EMT), drug resistance, and metastasis in NSCLC [67,68,69,70,71,72,73,74,75,76]. Moreover, the role of YAP in the cancer immunity of human NSCLC has recently been explored. Recent studies have found that YAP regulates tumor-associated immune cells in the tumor microenvironment and the expression of an immune checkpoint engaged by cancer cells to escape host antitumor immune responses [77,78,79,80,81,82]. 

The K-ras mutation is oncogenic and frequently occurs in NSCLC patients (15–30%), and there is still no approved effective targeted therapy for the clinical treatment of advanced NSCLC caused by K-ras mutations [83]. Currently, AMG 510 is the only drug with a potential antitumor effect on K-ras^G12C^-mutated metastatic NSCLC that has been investigated in a phase I clinical trial (NCT03600883) [84]. According to a study by Singh et al., some K-ras-mutated NSCLC cell lines, such as A549, H23, and SK-LU-1, contain K-ras-independent cells, and these cells do not require K-ras to maintain viability. This study demonstrated that inhibiting or knocking down K-ras has no suppression effect on the proliferation, migration, and invasive abilities of these K-ras-independent cells [85]. Two studies [67,68] found that some K-ras-mutated NSCLC cells relapse after K-ras exhaustion without the re-expression of the K-ras transgene. The YAP gene was found to be amplified in the relapsed NSCLC cells without the re-expression of K-ras, and inhibiting YAP suppressed cancer cell growth both in vitro and in vivo. In addition, YAP interacts with FOS to activate the mitogen-activated protein kinase (MAPK) pathway to induce the EMT in the absence of K-ras signaling [67,68]. Together, the findings of these studies suggest that YAP takes over K-ras as a driver of cancer when there is a loss of K-ras signaling in these K-ras-independent NSCLC cells.

EGFR and its downstream signaling Ras/Raf/MEK/ERK and PI3K/Akt/mTOR pathways are strongly associated with human NSCLC [86,87,88]. The Ras/Raf/MEK/ERK and PI3K/Akt/mTOR signaling pathways crosstalk with the Hippo/YAP signaling pathway and negatively regulate Hippo kinases to activate the oncogenic function of YAP in cancer cells [89,90,91,92]. For instance, Raf-1, the oncogenic protein of the MAPK and Akt pathways, was shown to regulate Hippo kinases, including mammalian sterile 20-like kinase 2 (MST2) and LATS1, in a previous study [93]. You et al. [90] showed that, in human NSCLC cells, the inhibition of ERK1/2 decreases YAP protein expression by accelerating protein degradation. The same study also found that YAP protein expression could be rescued during the knockdown of ERK1/2 depletion through forced overexpression of the ERK2 gene [90]. In addition, several studies found that the activation of YAP increases the downstream gene expression of EGFR ligands such as amphiregulin (AREG) and neuregulin 1 (NRG-1), as well as connective tissue growth factor (CTGF) and cysteine-rich angiogenic inducer 61 (CYR61), forming an autocrine loop to reinforce the MAPK signaling pathway in order to induce drug resistance and cancer metastasis [75,76,94,95,96,97]. Therefore, YAP has been suggested to be involved in the promotion of resistance to therapies targeting the EGFR/Ras/Raf/MEK/ERK pathway, and it represents a therapeutic target for metastatic NSCLC. In one clinical study, the overexpression of YAP in NSCLC with a BRAF V600E mutation led to a worse initial response to BRAF and MEK inhibitors [89]. One study showed that the inhibition of YAP synergizes the cytotoxicity of BRAF and MEK inhibitors to NSCLC cells containing BRAF and K-ras mutations [89]. Another study showed that the forced overexpression of YAP promotes resistance to EGFR-TKI erlotinib in the NSCLC cell line HCC827 (EGFR exon 19 deletion), and inhibiting YAP enhances the cytotoxicity of erlotinib to the NSCLC cell line H1975 (L858R + T790M mutations) [71]. One previous study also found that high tumor expression of YAP in NSCLC patients with the ALK fusion mutation was correlated with a poor response to ALK inhibitors [96]. The same study showed that inhibition of YAP re-enhanced the anti-tumor effect of ALK inhibitors in vitro and in vivo [96]. Studies investigating YAP-induced targeted therapy resistance are summarized in Table 2.

The role of YAP in promoting NSCLC metastasis has been investigated in vivo, and the inhibition of YAP has shown promising efficacy in suppressing NSCLC metastasis in mouse models [74,75,76]. Dubois et al. found that the tumor suppressor gene RASSF1A inhibits YAP through the GEF-H1/RhoB pathway, and they showed that RASSF1A depletion in a mouse model enhances the metastatic potential of the NSCLC cell line H1975 [74]. Another study found that the NSCLC cell lines H2030-BrM3 (K-ras^G12C^mutation) and PC9-BrM3 (EGFR^Δexon19^mutation) (derived from the NSCLC cell lines H2030 and PC9 in the laboratory of Professor Joan Massagué, Memorial Sloan Kettering Cancer Center, New York, NY, USA) [99,100], both of which have a high brain metastasis potential, had increased YAP expression levels compared with their parental cell lines (H2030 and PC9). This study also demonstrated that the genetic ablation of YAP by short hairpin RNA inhibits the brain metastatic ability of H2030-BrM3 cells in vivo [75]. Additionally, a recent study showed that a biochemical natural compound, cucurbitacin E (a tetracyclic triterpene isolated from cucurbitaceae), inhibits YAP protein expression in H2030-BrM3 and PC9-BrM3 cells. Cucurbitacin E was also shown to suppress the brain metastasis of H2030-BrM3 cells in a mouse model [76]. Oxidative stress was reported to promote the migration, invasion, and metastasis of NSCLC through the LATS2/YAP signaling pathway in a recent study [101]. Reactive oxygen species (ROS) are the downstream products of oxidative stress and have been identified to be involved in response to anti-cancer therapies in human NSCLC [102,103]. A previous study by Huang et al. reported that the activation of YAP increased the accumulation of ROS by downregulating the antioxidant enzyme GPX2 in human lung squamous cell carcinoma [104]. This study also found that small molecular digitoxin decreased the S127 phosphorylation of YAP and promoted the nuclear translocation of YAP. In addition, this study showed an anti-tumor effect of digitoxin in a lung squamous cell carcinoma PDX model with low YAP expression. The findings suggest that YAP has a tumor-suppressor function through downregulating GPX2 expression in a DNp63-dependent manner, and more studies focus on the regulation of ROS by YAP are needed to improve precision medicine for the treatment of lung squamous cell carcinoma [104].

Wingless/Ints (Wnt), bone morphogenetic proteins (BMPs)/transforming growth factor β (TGFβ), Notch, and Hedgehog (Hh) are oncogenic signaling pathways that are involved in the regulation of tumorigenesis in NSCLC [99,105,106,107,108]. Wnt/β-catenin was found to physically activate YAP signaling in HEK293 cells in a previous study [109]. Another recent study showed that the concurrent activation of Wnt/β-catenin and YAP signaling promotes cancer progression and is associated with poor prognosis in NSCLC [110]. A previous study demonstrated that YAP forms a positive feedback loop with the Notch and Wnt/β-catenin signaling pathways to promote liver tumorigenesis, and Hippo kinases repress this loop [111]. Under normal physiological conditions, the interactions of TGFβ, Hh, and YAP play a critical role in lung development and vascular formation and are also involved in the regulation of pulmonary diseases such as fibrosis and emphysema [112,113]. To date, the roles of interactions among TGFβ, Hedgehog, and YAP have not been clarified and more studies are needed to explore them. 

The regulation of the Hippo kinase cascade, YAP, and other signaling pathways is complex. Previous studies have shown that signaling pathways, including MAPK, PI3K, Wnt, TGFβ, Notch, and Hh, directly activate YAP to form a positive loop in part and partly activate YAP by repressing Hippo kinases [89,90,91,92,93,94,95,96,109,110,111,112,113]. Loss of function or mutations in Hippo kinases, including NF2 and LATS1/2, lead to the activation of YAP, and then YAP activation positively interacts with other signaling pathways to promote cancer progression [114,115,116]. Contrarily, it has been reported that tumor suppressors, including RASSF1A and p53, negatively regulate YAP by activating the Hippo kinase cascade. The loss of function mutations of RASSF1A or p53, contributes to increased YAP expression in cancers [74,117]. 

In cellular assays, the Hippo/YAP pathway is physiologically critical for promoting the proliferation and regulation of anti-apoptosis in normal cells such as cardiomyocytes and endometrial stromal cells [118,119]. Two previous studies demonstrated that YAP-induced anti-apoptosis worsens the initial treatment response to BRAF and MEK inhibitors and EGFR-TKIs in NSCLC cells with K-ras, BRAF-V600E, or EGFR mutations [89,97]. The two studies showed that the inhibition of YAP by either genic or pharmacological ablation restores the cytotoxicity of BRAF and MEK inhibitors and EGFR-TKIs to NSCLC cells [89,97]. The findings of the two studies indicate that YAP-induced proliferation and anti-apoptosis are important factors in treatment resistance in NSCLC.

Taken together, these studies provide evidence that therapies targeting YAP alter drug resistance in NSCLC and effectively suppress the migration, invasion, and metastasis of human NSCLC in vitro and in vivo. Further development and investigation of drugs targeting YAP for the treatment of metastasis are warranted.

The regulation of the Hippo/YAP pathway in NSCLC is summarized in Figure 3.

## 4. Crosstalk between the Src and Hippo/YAP Signaling Pathways

Src protein kinase and Hippo/YAP can coexist in normal organ tissues, and the interaction of SRC and YAP is crucial for maintaining physiological function in normal cells [120,121]. For example, a study found that gp130 (a co-receptor of IL-6 cytokines) induces the phosphorylation of YAP by cell surface receptors through the activation of the tyrosine kinases Src and Yes [121]. This phosphorylation of YAP increases the stabilization and nuclear translocation of YAP, which activates downstream growth factor genes to promote tissue growth and regeneration during intestinal mucosal injury [121]. The crosstalk between the Src and Hippo/YAP signaling pathways is also involved in the promotion of human diseases, including fibrosis and inflammatory disorders. In fibroblast cells, actomyosin regulates the accumulation of nuclear YAP through the Src family kinase [122,123]. A study demonstrated that nuclear export is a key mechanism for determining YAP localization, and XPO1 (Exportin 1) induces YAP nuclear export by serine phosphorylation. This study found that the SRC family kinase regulates the accumulation of nuclear YAP through the tyrosine phosphorylation of YAP and inhibits XPO1 to decrease YAP nuclear export in fibroblast cells [122]. Two recent studies reported that crosstalk between the Src and Hippo/YAP pathways is involved in the pathophysiology of renal fibrosis [124,125]. One of the two studies showed that Rac-GTPases promote the fibrotic TGF-β1 pathway through the Src-mediated Hippo/YAP pathway using a chronic kidney disease mouse model. In addition, they found that the inhibition of Rac reduces Src and YAP expression levels and attenuates the progression of renal fibrosis in this mouse model [124]. The findings of the other study demonstrated that the farnesoid X receptor (FXR) protects against renal fibrosis by downregulating the Src-mediated Hippo/YAP pathway. Additionally, they showed that in the kidneys of FXR knockout mice, there is increased expression of the transcriptional downstream genes of YAP, fibrosis markers, and inflammatory genes when compared with those in wild-type mice [125]. The interaction of the Src kinase family and YAP is crucial for the pathological activation of cancer-associated fibroblasts, and the accumulation of nuclear YAP increases the expression of the downstream genes required for pro-tumorigenic functions [122,126]. Similar findings were shown in another earlier study, which indicated that the activation of YAP by Src is required for the generation and maintenance of cancer-associated fibroblasts [127]. Therefore, several previous studies have been engaged in the investigation and development of anticancer therapies that target the Src-mediate Hippo/YAP signaling pathway in various cancers. 

In cancer cells, transmembrane cell surface growth factor receptors and downstream intracellular signaling pathways, including EGFR, PDGFR, PI3K/Akt/mTOR, and RhoA, have been reported to mainly activate YAP mediated by Src kinase through three mechanisms: (1) direct phosphorylation; (2) the activation of pathways repressing Hippo kinases; and (3) Hippo-independent mechanisms [128,129,130,131,132,133,134,135]. The Src kinase YES1 phosphorylates YAP at the site of tyrosine 357 (Y357) to activate YAP, and Y357 phosphorylation of YAP is required for Wnt/β-catenin signaling to maintain survival and tumorigenesis in human colorectal cancer cells [129]. The Y357 phosphorylation of YAP by YES1 induces the expression of the transcriptional genes BCL2L1 and BIRC5 downstream of YAP, and the small-molecule YES1 inhibitor has been found to suppress the proliferation of β-catenin-dependent cancers in cell lines and in vivo experiments [129]. The activation of YAP through the phosphorylation of integrin-Src signaling is crucial for controlling skin homeostasis [130]. In addition, a study demonstrated that concurrent tyrosine phosphorylation at sites 341, 357, and 394 by Src kinases is essential for YAP transcriptional activity, nuclear localization, and interaction with TEAD in skin squamous cell carcinomas [131]. Another study showed that PDGFR upregulates YAP transcriptional activity via Y357 phosphorylation mediated by Src kinases in cholangiocarcinoma [132]. 

Several previous studies have provided evidence indicating that Src represses the Hippo kinase LATS by tyrosine phosphorylation, which induces the transcriptional activity of YAP, promoting tumorigenesis and cancer metastasis [126,128,129,130,131,132,133,134,135,136,137,138,139]. GPCR-kinase-interacting protein 1 (GIT1) promotes the LATS-mediated phosphorylation of YAP, which leads to cytoplasmic retention and the degradation of YAP [128,133]. Lamar et al. found that Src also inhibits GIT1, reducing the LATS-mediated phosphorylation of YAP in several cancer cells [128,133]. One previous study showed that, in human breast cancer tissues, the accumulation of active nuclear dephosphorylated YAP on the LATS1 target site, as assayed by immunohistochemistry (IHC) staining, was correlated with Src protein levels [134]. Ando et al. found that the tissue inhibitor of metalloproteinase-1 (TIMP-1) activates YAP by suppressing LATS to promote cancer cell proliferation. In the same study, they concluded that TIMP-1 activates Src, promoting downstream RhoA-mediated F-actin assembly, leading to LATS inactivation [135]. Another study showed that macrophage-associated immunosuppression regulates angiogenesis via Src-PI3K-YAP signaling in glioblastoma [138]. In fact, many studies have demonstrated that Src downstream signaling, including through the MAPK and PI3K pathways, negatively regulates Hippo kinases, thereby activating YAP [90,91,95,96,97,126,127,128,129,130,131,132,133,134,135,136,137,138,139]. Collectively, the findings of these studies indicate that Src can influence the activation of YAP through the repression of Hippo kinases, either by direct inhibition or through interaction with its upstream cell surface receptors and downstream intracellular cell signaling pathways. 

In some rare conditions, Src influences YAP activity through mechanisms other than the direct phosphorylation of YAP and the repression of Hippo kinases [140]. For instance, Hu et al. demonstrated that integrin α3 uses the FAK/Src-CDC42-PP1A signaling cascade to regulate YAP-S397 phosphorylation and nuclear localization in transit-amplifying cells [140]. In the same study, they concluded that the activation of YAP through FAK/SRC-CDC42-PP1A induces mTOR signaling, promoting transit-amplifying cell proliferation [140]. Future studies may be needed to unravel the other mechanisms of SRC-mediated regulation of the Hippo/YAP pathway, i.e., aside from the direct phosphorylation and suppression of Hippo kinases. 

Recently, two studies identified that the Src–YAP signaling axis is associated with resistance to targeted therapy in human NSCLC [40,97,141]. In a previous study [40] conducted by Fan et al., acquired YES1 amplification was detected in five EGFR mutant NSCLC patients (three had the L858R mutation and two had the exon 19 deletion mutation) who were pre-treated with the EGFR-TKIs erlotinib or afatinib and had acquired resistance to EGFR-TKIs [40]. One study by Chaib et al. found that EGFR mutant lung cancer cells survive initial EGFR-TKI therapy through the co-activation of STAT3 and Src–YAP signaling. In a cohort analysis of 64 EGFR mutant NSCLC patients (62 had common mutations L858R and exon 19 deletion; the other two had the uncommon mutations L861Q and G719X) treated with first-line EGFR-TKIs, patients with a high level of expression of STAT3 or YAP1 had a worse progression-free survival following EGFR-TKI therapy. Through in vitro and in vivo experiments, the same study showed that treatment with the EGFR-TKI gefitinib, in combination with the SRC inhibitor saracatinib (AZD05300), had a synergistic antitumor effect on NSCLC cells [97]. Taken together, Src–YAP signaling appears to give primary and acquired resistance to EGFR-TKIs in NSCLC patients with sensitive EGFR mutations (L858R and exon 19 deletions). The other study demonstrated that the combination of the Src family kinase inhibitor dasatinib and the MEK inhibitor trametinib decreased YAP protein expression in K-ras mutant NSCLC cells. This combination also demonstrated a synergistic effect by suppressing NSCLC tumor growth in xenograft mouse models [142]. EGFR-TKIs in combination with a therapy targeting the Src–YAP axis may overcome the resistance to EGFR-TKIs, and future clinical trials investigating this combination in EGFR-mutant NSCLC patients are warranted. Another previous analysis suggested that the Src pathway is associated with high metastatic potential in human NSCLC cells [141]. Therefore, future investigations and the development of new therapies targeting the Src–YAP axis to treat advanced NSCLC are warranted. The crosstalk between the Src and Hippo/YAP signaling pathways is summarized in Figure 4.

## 5. Future Perspectives: Potential Therapies Targeting Src–YAP in NSCLC

### 5.1. Src Inhibitors—Dasatinib, Bosutinib, and Saracatinib

Dasatinib and bosutinib are the two Src inhibitors that are currently approved by the US Food and Drug Administration (FDA) for the treatment of hematological malignancies, including chronic myelogenous leukemia (CML) and acute lymphoblastic leukemia (ALL) [143,144,145]. Several previous studies have shown the effect of dasatinib in inhibiting YAP expression in various cancer cells, including renal cell carcinoma, breast cancer, and NSCLC [143,146,147]. The efficacy and safety of dasatinib has been investigated in combination with erlotinib in relation to the treatment of NSLCC patients in a phase I/II clinical trial (ClinicalTrials.gov NCT00826449). The results of this clinical trial concluded that dasatinib combined with erlotinib is safe and feasible for use in NSCLC. However, the same study did not suggest using this combination in molecularly unselected NSCLC [148]. Another Src inhibitor, saracatinib (AZD0530), was investigated in a phase II clinical trial for previously treated advanced NSCLC patients (ClinicalTrials.gov NCT00638937) [149]. Only a subset of saracatinib-responsive NSCLC patients was shown in the study, but the saracatinib-responsive NSCLC patients were not molecularly defined [149]. In a recent preclinical study, dasatinib has demonstrated to have an antitumor effect in NSCLC PDX models with high YES1 expression [56]. The findings of this study suggested that dasatinib is a therapy that could potentially benefit selected NSCLC patients with high YES1 expression or genetic amplification [56]. In addition, YES1 amplification was detected in NSCLC patients who had acquired resistance to EGFR or ALK inhibitors [40]. Therefore, the combination of dasatinib and targeted therapies in NSCLC patients with known targetable driver mutations may delay the occurrence of acquired resistance and even overcome acquired resistance in molecularly selected patients [40]. A recent study reported that the DNA methyltransferase inhibitor upregulates the Hippo-activators RASSF1 and RASSF5 through promoter demethylation in human rhabdomyosarcoma cells [150]. The DNA methyltransferase inhibitor decreases the activity of YAP through the activation of canonical Hippo kinases and phosphorylation of YAP at the serine 127 site [150]. The same study demonstrated that the combination of DNA methyltransferase inhibitor and dasatinib had a synergic anti-tumor effect in treating rhabdomyosarcoma in vitro and in vivo [150]. Dasatinib is an Src–YAP inhibitor for which future clinical studies may be conducted to investigate its efficacy in treating selected NSCLC patients. 

### 5.2. Inhibition of Src Downstream Intracellular Signaling Pathways

Small-molecule compounds that inhibit intracellular signaling pathways, including MAPK, PI3K, Rho/ROCK, and STAT3, have been developed and are under investigation in relation to the treatment of NSCLC. An MEK inhibitor, trametinib, has been widely used in the clinical treatment of melanoma, thyroid cancer, and NSCLC [151]. The dual therapy of trametinib combined with dabrafenib (BRAF inhibitor) is effective for treating metastatic NSCLC with a BRAFV600E mutation [151]. In BRAF mutant NSCLC, dual therapy (trametinib + dabrafenib) has a higher response rate (~60%) than single therapy with dabrafenib or vemurafenib alone (30–40% response rate) [152,153]. The combination of trametinib with chemotherapy or other targeted therapies is currently under investigation in many preclinical and clinical studies [154,155]. While previous studies have shown that small-molecule drugs inhibiting PI3K, Rho/ROCK, and STAT3 downregulate Src–YAP expression, these studies have been limited to preclinical investigations [63,142,156,157,158]. 

### 5.3. Disruption of the YAP–TEAD Complex

Verteporfin, a benzoporphyrin derivative, is used clinically as a photosensitizer of photodynamic therapy to treat macular degeneration and central serous retinopathy [159]. Verteporfin blocks the formation of the YAP–TEAD complex and inhibits the YAP-related oncogenic effect both in vitro and in animal model experiments [158,159,160,161]. While verteporfin has been shown to be a good YAP inhibitor in previous studies on NSCLC, its use has been limited to preclinical models [158,159,160,161]. In fact, verteporfin is only optimized for photodynamic therapy to treat vascular disorders of the eyes and is not designed for clinical anticancer therapy [162,163]. A recent study showed that the reactivation of YAP promotes survival and dormancy in EGFR mutant NSCLC cells, while EGFR downstream signaling is suppressed by EGFR and MEK inhibitors [98]. Two YAP–TEAD inhibitors, XAV939 and MYF-01-37, were investigated in the same study, and both YAP–TEAD inhibitors enhanced EGFR inhibitor-mediated apoptosis and prevented dormancy [98]. Further studies to investigate the efficacy of XAV939 and MYF-01-37 in treating NSCLC are warranted. 

### 5.4. Cyclin-Dependent Kinase 1,7,9

The cyclin-dependent kinase (CDK) family consists of protein kinases that play an important role in regulating the cell cycle. Some CDKs have been found to function in the phosphorylation of YAP and the regulation of transcriptional genes downstream of YAP [164,165,166,167,168,169,170,171]. CDK1 has been shown to phosphorylate YAP at sites T119 and S289 during the G2/M phase of the cell cycle, promoting cell migration and invasion in cancer cells [164,165,166]. CDK7 has been reported to regulate the phosphorylation of RNA polymerase II and transcription factors to maintain an oncogenic state in various cancers, including breast and lung cancers, neuroblastoma, and leukemia [167]. Another study demonstrated that the inhibition of CDK7 attenuates the YAP protein expression level through cytoplasmic retention and degradation [168]. A CDK7 small-molecule inhibitor, THZ1, has been developed and shown to have effects in interfering with the transcriptional function of CDK7 and the activation of YAP [167,168]. Recently, Cho et al. found that CDK7 directly regulates YAP through phosphorylation at S129 [169]. Therefore, CDK7 may also be a potential YAP inhibitor, and the combination of Src with CDK7 inhibitors has been suggested as an option for inhibiting YAP activity in cancers. CDK9 was reported to play a key role in mediating the elongation complex of transcriptional activation driven by the YAP–TEAD complex [170]. CDK9 inhibitors, including Flavopiridol (alvocidib), dinaciclib, seliciclib, and MC180295, have been developed and evaluated in preclinical studies [170,171,172,173]. Zhang et al. recently developed a highly selective CDK9 inhibitor, MC180295, and showed that MC180295 not only has an anticancer effect but that it is also an effective anti-PD-1 immune checkpoint inhibitor, as demonstrated through both in vitro and in vivo experiments [171]. Drugs targeting the CDK family have been suggested as potential therapies for cancers with high YAP activation, and further investigation through future studies is warranted. 

The potential future therapies targeting Src–YAP for the treatment of NSCLC are labeled in Figure 5. Chemical compounds and drugs targeting the Src–YAP axis are summarized in Table 3. 

## 6. Conclusions

Our review indicates that Src-mediated Hippo/YAP pathways play important roles in promoting cancer progression, metastasis, and drug resistance in NSCLC. The inhibition of multiple points of the Src–YAP axis is suggested to be a potential therapeutic strategy for advanced NSCLC, based on the findings of previous studies. Future work should focus on determining the efficacy of small-molecule drugs that may potentially inhibit Src–YAP and even combination therapies for advanced NSCLC.

## Figures and Tables

**Figure 1 cancers-12-01361-f001:**
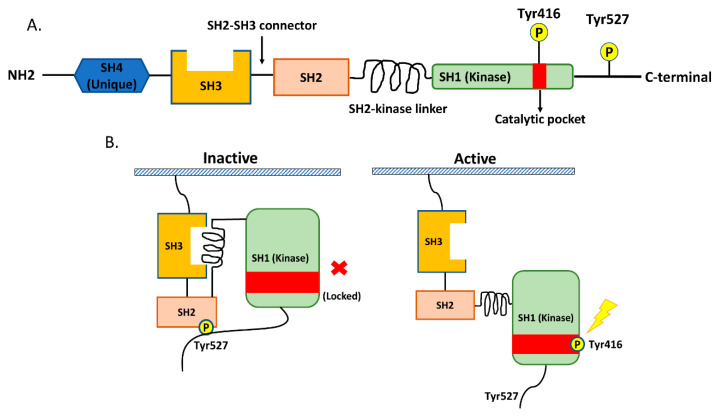
The structure of Src and the regulation of its kinase activity. (**A**) Src consists of the SH1 (tyrosine kinase), SH2, SH3, and SH4 (unique) domains; the SH3–SH2 connector; the SH2–kinase linker; a C-terminal tail regulatory region; and two tyrosine sites (Tyr416 and Tyr527). (**B**) The regulation of Src activity depends on the phosphorylation of two tyrosine sites and intramolecular interactions among the domains. Normally, phosphorylated Tyr527 binds to the SH2 domain and the SH2–kinase linker binds to the SH3 domain, and these binding events result in the protection of the catalytic pocket of Tyr 416 in the kinase domain (SH1) from inappropriate phosphorylation. The dephosphorylation of Ty527 results in conformational change, unlocking the catalytic pocket of Tyr 416 and subsequently activating Src through the intramolecular autophosphorylation of Tyr416.

**Figure 2 cancers-12-01361-f002:**
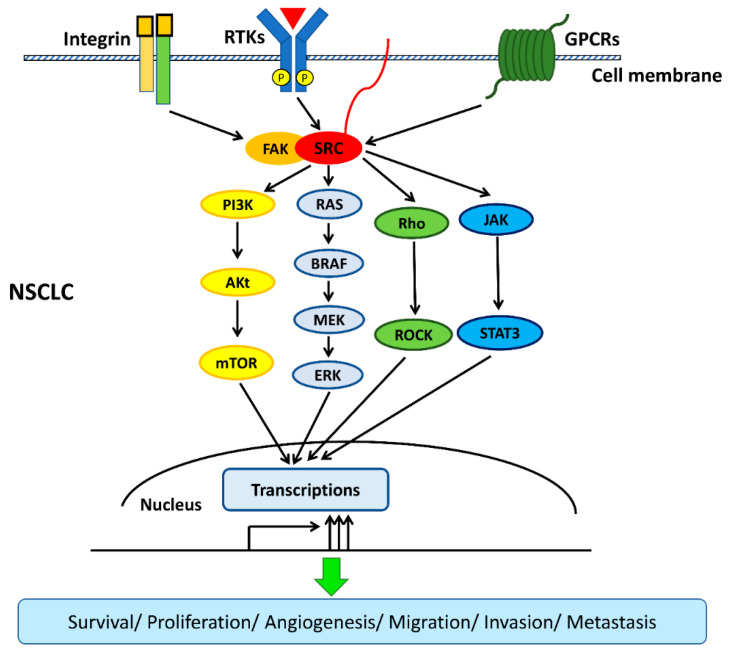
Src interacts with transmembrane cell surface receptors and mediates the transduction of intracellular downstream signaling pathways. Src interacts with transmembrane cell surface receptors, including the integrin/focal adhesion kinase (FAK), the receptor of tyrosine kinases (RTKs), and G-protein-coupled receptors (GPCRs), and then activates downstream signaling pathways. Src activates downstream signaling pathways, including mitogen-activated protein kinase (MAPK)/extracellular signal-regulated kinase (ERK), phosphatidylinositol 3-kinase (PI3K)/Akt/mTOR, interleukin 6 (IL-6)/Janus kinase (JAK)/signal transducer and the activator of transcription 3 (STAT3), and Rho/Rho-associated protein kinase (ROCK) pathways to promote survival, proliferation, angiogenesis, migration, invasion, and metastasis in non-small-cell lung cancer (NSCLC).

**Figure 3 cancers-12-01361-f003:**
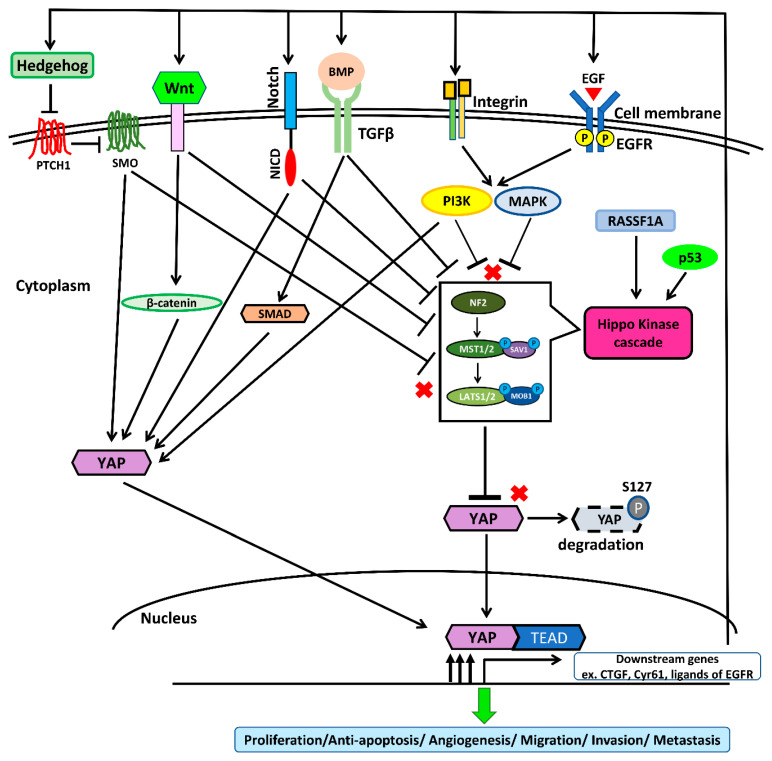
The regulation of the Hippo/YAP signaling pathway in non-small-cell lung cancer (NSCLC). The Hippo kinase cascade consists of neurofibromatosis 2 (NF2), mammalian sterile 20-like kinase 1/2 (MST 1/2), and large tumor suppressor homolog 1/2 (LATS1/2). In the cytoplasm, Hippo kinases phosphorylate the Yes-associated protein (YAP) at serine 127, leading to the sequestration and degradation of YAP. Tumor suppressors, including RASSF1A and p53, activate the function of Hippo kinases. Signaling pathways, including MAPK, PI3K, Wnt, TGFβ, Notch, and Hh, directly activate YAP to form a positive loop, in part, and partly activate YAP by repressing Hippo kinases. The activation of transcriptional downstream genes, including CTGF, CYR61, and other ligands of the epidermal growth factor receptor (EGFR), leads to the formation of positive autocrine loops that activate oncogenic pathways. The formation of these autocrine loops enhances YAP signaling, which promotes tumor cell proliferation, anti-apoptosis, angiogenesis, migration, invasion, and metastasis in NSCLC.

**Figure 4 cancers-12-01361-f004:**
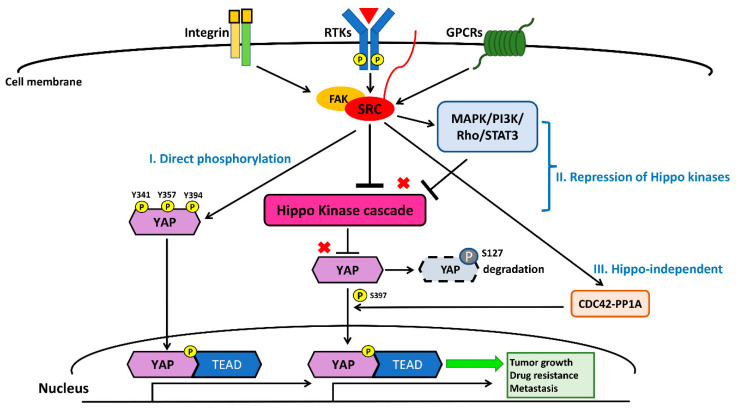
Crosstalk between the Src and Hippo/YAP signaling pathways. Src activates YAP through three main mechanisms: (1) direct phosphorylation at sites Y341, Y357, and Y394 leads to the nuclear translocation of YAP; (2) the repression of Hippo kinases by direct inhibition and/or the activation of pathways, such as MAPK, PI3K, Rho, and STATs, suppresses Hippo kinases; and (3) a Hippo-independent mechanism.

**Figure 5 cancers-12-01361-f005:**
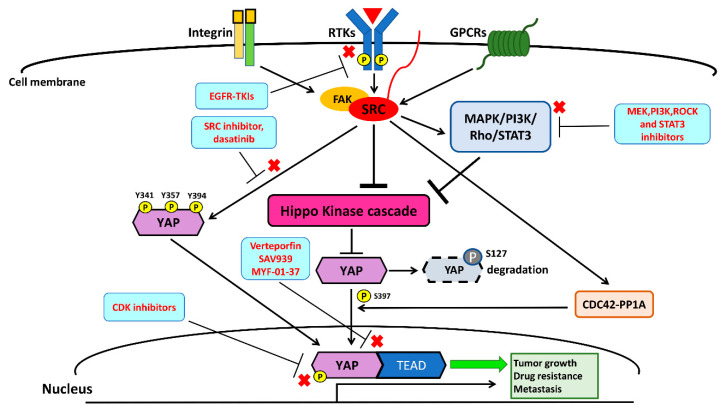
Src–YAP blockade by EGFR-TKIs, MAPK, PI3K, Rho/ROCK, and STAT3 inhibitors, Src inhibitor, YAP–TEAD complex inhibitor, and cyclin-dependent kinase (CDK) 1, 7, and 9 inhibitors, which could be potential future targets in therapies for the treatment of NSCLC.

**Table 1 cancers-12-01361-t001:** Src-altered resistance to targeted therapy in non-small-cell lung cancer (NSCLC).

Genomic Alternation of Src	Primary Mutation of NSCLC	Resistance to Target Therapy (Primary/Secondary)	Targeted Therapy Resistance	Type of Study	Reference
Overexpression of SHP2	EGFR mutation L858R/exon 19 deletion/G719X/L861Q	Primary	Erlotinib, Gefitinib, Afatinib	Clinical analysis	[39]
YES1 amplification	EGFR mutation L858R/exon 19 deletion	Secondary	Erlotinib, Afatinib	Clinical analysis	[37]
YES1 amplification	EGFR mutation Exon 19 deletion/L858R+T790M	Secondary	Osimertinib	Preclinical study	[38]
YES1 amplification	EGFR mutation L858R/exon 19 deletion ALK fusion EML4-ALK fusion/HIP1-ALK fusion	Secondary	Erlotinib, Afatinib, Crizotinib, Ceritinib	Clinical analysis	[40]

ALK: anaplastic lymphoma kinase; EGFR: epidermal growth factor receptor; EML4: echinoderm microtubule-associated protein-like 4.

**Table 2 cancers-12-01361-t002:** YAP-related resistance to targeted therapy in NSCLC.

YAP Alternation	Primary NSCLC Mutation	Resistance to Target Therapy (Primary/Secondary)	Targeted Therapy Resistance	Type of Study	Reference
YAP overexpression	EGFR mutation Exon 19 deletion/L858R+T790M	Primary/secondary	Erlotinib	Preclinical study	[71]
YAP overexpression	EGFR mutation L858R/exon 19 deletion/T790M	Primary/secondary	Erlotinib, Gefitinib, Afatinib, Osimertinib	Preclinical study and clinical analysis	[94]
YAP overexpression	L858R/exon 19 deletion /G719X/L861Q	Primary	Erlotinib, Gefitinib, Afatinib, Icotinib	Preclinical study and clinical analysis	[97]
YAP overexpression	EGFR mutation L858R/exon 19 deletion/T790M	Secondary	Osimertinib, Trametinib	Preclinical study	[98]
YAP overexpression	ALK fusion EML4-ALK fusion/	Secondary	Crizotinib, Ceritinib	Preclinical study	[96]
YAP overexpression	BRAF V600E, K-ras	Primary	Vermurafenib, Trametinib	Preclinical study	[89]

**Table 3 cancers-12-01361-t003:** Chemical compounds and drugs targeting the Src–YAP axis.

Target	Drug	Status in NSCLC	Reference
**Src**	Dasatinib	Phase I/II clinical trial (Completed) NCT00826449	[148]
	Bosutinib	Phase I clinical trial (Recruiting) NCT03023319	[41]
	Saracatinib (AZD0530)	Phase II clinical trial (Completed) NCT00638937	[149]
**MAPK**	Trametinib	Combined with dabrafenib in BRAF V600E mutation NSCLC (Approved by US FDA)	[152,153]
**PI3K**	Taselisib	Preclinical study	[156]
**STAT3**	TPCA-1	Preclinical study	[97]
**Rho/ROCK**	GSK269962A	Preclinical study	[63,158]
**YAP–TEAD**	Verteporfin	Preclinical study	[159,160,161]
	XAV939	Preclinical study	[98]
	MYF-01-37	Preclinical study	[98]
**CDK1,5,9**	Dinaciclib (MK7965)	Phase II clinical trial (Completed) NCT00732810	[170,172]
**CDK7**	THZ1	Preclinical study	[167,168,169]
**CDK9**	Seliciclib	Phase II clinical trial (Terminated) NCT00372073	[170,173]
	Flavopiridol (alvocidib)	Phase I clinical trial (Terminated) NCT00094978	[170,171,173]
	MC180295	Preclinical study	[170,171]

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
