# Peer review of "The Crosstalk between Src and Hippo/YAP Signaling Pathways in Non-Small Cell Lung Cancer (NSCLC)"

_cancers, 2020, doi:10.3390/cancers12061361_

Round 1

Reviewer 1 Report

The paper is fluently and clear.

The subject is interesting and even the figure are clear.

Author Response

Point 1: English language and style are fine/minor spell check required ʉ۬

Response 1: The manuscript has been revised by English language editing service of MDPI.

Point 2: The paper is fluently and clear. The subject is interesting and even the figure are clear. 

Response 2: We appreciate the reviewer’s enthusiastic comments about our review article.

Reviewer 2 Report

Hsu and colleagues comprehensively reviewed the current studies on the correlation between Src and Hippo/Yap pathways in terms of their roles in NSCLC. At the start, the authors summarize the research on both Src and Hippo/Yap signaling individually in an NSCLC and then their crosstalk, including clinical related progress. In the end, they focused on the potential therapies targeting Src-Yap in NSCLC. The review is most updated and comprehensive. It captures the readers’ interest for the journal.

Author Response

Response to Reviewer 2 Comments

Hsu and colleagues comprehensively reviewed the current studies on the correlation between Src and Hippo/Yap pathways in terms of their roles in NSCLC. At the start, the authors summarize the research on both Src and Hippo/Yap signaling individually in an NSCLC and then their crosstalk, including clinical related progress. In the end, they focused on the potential therapies targeting Src-Yap in NSCLC. The review is most updated and comprehensive. It captures the readers’ interest for the journal.

Point 1: Moderate English changes required ʉ۬

Response 1: The manuscript has been revised by English language editing service of MDPI.

Point 2: The review is most updated and comprehensive. It captures the readers’ interest for the journal.

Response 2: We appreciate the reviewer’s enthusiastic comments about our research article.

Reviewer 3 Report

In this review, Hsu and colleagues focus their attention on the regulation of yes-associated protein (YAP)/Hippo activation signaling pathways promoted by proto-oncogene tyrosine-protein kinase (Src) and discuss the key open questions about the present and future therapy targeting Src-YAP for the treatment of NSCLC.

In my opinion, the paper is interesting and suitable for publication by considering the structure and purpose of the review. The authors summarize and address the rationale about the crosstalk between Src family kinases and the Hippo/YAP signaling pathway in a clear and comprehensible way. The scientific content is good and the captions are really appreciated because describe and resume the transduction of intra/extracellular signaling concerning Src and Hippo/YAP signaling pathways and their related therapeutic targets. The English language used in the manuscript is adequate and respect the guidelines of our journal.

By consequence, there are some minor comments to be taken into consideration as indicated below.

  1. YAP and TAZ, the major effectors of the Hippo pathway, are fully studied since the last decade and contribute in the regulation of the phosphorylation-induced cytoplasmic retention and protein degradation in response to intrinsic and extrinsic signals. So there are a large number of proteins that regulate YAP localization or transactivation. I wonder if the upstream signals and the peripheral Hippo pathway components that relay signals to the core kinase cascade may cross-talk with Wingless/Ints (Wnt), bone morphogenetic proteins (BMPs), Notch, and Hedgehog (Hh) also in NSCLC. Please deepen your explanation through the text.
  2. It is certain that YAP activates the Hippo pathway in a negative feedback. I suggest to the authors to create a subparagraph in the section regarding the crosstalk between Hippo/YAP signaling and other pathways in order to distinguish the negative and positive loop regulation in which are involved.
  3. As described in the paper, Hippo signaling is sensitive to biochemical cues mediated through protein-protein interactions at cell-cell or cell-ECM contacts. In addition, the activation of this pathway surely depends on mechanical stress. Based on this assumption, it is well known that several stress signals can modulate YAP and TAZ activities. Specifically, hypoxia seems to induce YAP and TAZ activation by inhibiting LATS. Is there any evidence about the relationship between the role of YAP and reactive oxygen species (ROS) in NSCLC? Please explain it.
  4. An important function of the Hippo pathway seems to be the inactivation of YAP and TAZ in differentiated cells to maintain cell quiescence. YAP and TAZ are activated to promote stem/progenitor cell self-renewal and tissue after Hippo pathway suppression. Taking into the consideration that these effectors are implicated in cancer initiation and development, what do we know about Hippo pathway and cellular assays such as apoptosis and proliferation? I think that this section needs to be implemented in light of therapeutic solutions in NSCLC.

Author Response

Response to Reviewer 3 Comments

In this review, Hsu and colleagues focus their attention on the regulation of yes-associated protein (YAP)/Hippo activation signaling pathways promoted by proto-oncogene tyrosine-protein kinase (Src) and discuss the key open questions about the present and future therapy targeting Src-YAP for the treatment of NSCLC.

In my opinion, the paper is interesting and suitable for publication by considering the structure and purpose of the review. The authors summarize and address the rationale about the crosstalk between Src family kinases and the Hippo/YAP signaling pathway in a clear and comprehensible way. The scientific content is good and the captions are really appreciated because describe and resume the transduction of intra/extracellular signaling concerning Src and Hippo/YAP signaling pathways and their related therapeutic targets. The English language used in the manuscript is adequate and respect the guidelines of our journal.

By consequence, there are some minor comments to be taken into consideration as indicated below.

Point 1: English language and style are fine/minor spell check required.

Response 1: The manuscript has been revised by English language editing service of MDPI.

.

Point 2: YAP and TAZ, the major effectors of the Hippo pathway, are fully studied since the last decade and contribute in the regulation of the phosphorylation-induced cytoplasmic retention and protein degradation in response to intrinsic and extrinsic signals. So there are a large number of proteins that regulate YAP localization or transactivation. I wonder if the upstream signals and the peripheral Hippo pathway components that relay signals to the core kinase cascade may cross-talk with Wingless/Ints (Wnt), bone morphogenetic proteins (BMPs), Notch, and Hedgehog (Hh) also in NSCLC. Please deepen your explanation through the text.

Response 2: We have added a paragraph to discuss about Hippo/YAP cross-talks with Wingless/Ints (Wnt), bone morphogenetic proteins (BMPs), Notch, and Hedgehog (Hh) also in NSCLC as suggested.

In addition, we included these pathways in the figure 3. to address this comment.

Wingless/Ints (Wnt), bone morphogenetic proteins (BMPs)/ transforming growth factor β (TGFβ), Notch, and Hedgehog (Hh) are oncogenic signaling pathways which are involved in regulating tumorigenesis in NSCLC [1-6]. Wnt/β-catenin was found that physically activates YAP signaling in HEK293 cells in a previous study. Another recent study showed the concurrent activation of Wnt/β-catenin and YAP signaling promotes cancer progression and is associated with poor prognosis in NSCLC [7,8]. A previous study demonstrated that YAP forms a positive feedback loop with Notch and Wnt/β-catenin signaling pathways to promote liver tumorigenesis, and Hippo kinases repress this loop [9]. In normal physiological manner, the interaction of TGFβ, Hedgehog and YAP plays critical role in lung development and vascular formation, and is also involved in the regulation of pulmonary diseases such as fibrosis and emphysema [10,11]. To date, the role of interaction between TGFβ, Hedgehog and YAP is not clear and more studies needed to explore it.

This discussion paragraph was added in chapter 3, page 9 line 277-278 in revised manuscript

Point 3: It is certain that YAP activates the Hippo pathway in a negative feedback. I suggest to the authors to create a subparagraph in the section regarding the crosstalk between Hippo/YAP signaling and other pathways in order to distinguish the negative and positive loop regulation in which are involved.

Response 3: In response to point of Hippo/YAP signaling and other pathways in order to distinguish the negative and positive loop regulation in which are involved, we had added a paragraph to discuss about it in revised manuscript.

We also re-edited figure 3 with including the negative and positive loop regulation of Hippo/YAP to address this comment.

The regulation of Hippo kinase cascade, YAP and other signaling pathways is complex. Previous studies have shown that the signaling pathways including MAPK, PI3K, Wnt, TGFβ, Notch, and Hh directly activate YAP to form a positive loop in part, and partly activate YAP by repressing Hippo kinases [12]. Loss of function or mutations in Hippo kinases including NF2 and LATS1/2 lead to the activation of YAP, and then YAP activation positively interacts with other signaling pathways to promote cancer progression [12,13]. Contrarily, tumor suppressors including RASSF1A and p53 negatively regulate YAP by activating Hippo kinase cascade, and loss function of RASSF1A or p53 contributing to increased YAP expression in cancers had been reported [14,15].

This discussion paragraph was added in chapter 3, page 9 line 289-297 in revised manuscript

Point 4:  As described in the paper, Hippo signaling is sensitive to biochemical cues mediated through protein-protein interactions at cell-cell or cell-ECM contacts. In addition, the activation of this pathway surely depends on mechanical stress. Based on this assumption, it is well known that several stress signals can modulate YAP and TAZ activities. Specifically, hypoxia seems to induce YAP and TAZ activation by inhibiting LATS. Is there any evidence about the relationship between the role of YAP and reactive oxygen species (ROS) in NSCLC? Please explain it.

Response 4: In response to the point about “any evidence about the relationship between the role of YAP and reactive oxygen species (ROS) in NSCLC”, YAP increased accumulation of ROS by downregulating the antioxidant enzyme GPX2 in human lung squamous cell carcinoma has been reported in a previous study [16].

Oxidative stress was reported to promote migration, invasion and metastasis of NSCLC through LATS2/YAP signaling pathway in a recent study [16-19]. Reactive oxygen species (ROS) are the downstream productions of oxidative stress and have been identified in human NSCLC involved in response to anti-cancer therapies [16-19]. A previous study of Huang et al. reported that the activation of YAP increased accumulation of ROS by downregulating the antioxidant enzyme GPX2 in human lung squamous cell carcinoma [16-19]. This study also found that small molecular digitoxin decreased S127 phosphorylation of YAP and promoted nuclear translocation of YAP. In addition, this study showed that digitoxin had anti-tumor effect in a lung squamous cell carcinoma PDX model with low YAP expression. The findings suggested a tumor-suppressor function of YAP through downregulating GPX2 expression in DNp63-dependent manner, and more studies focus on YAP regulating ROS needed to improve precision medicine of lung squamous cell carcinoma [16-19].

This discussion paragraph was added in chapter 3, page 8 line 265-270, page 9 line 271-276 in revised manuscript

Point 5: An important function of the Hippo pathway seems to be the inactivation of YAP and TAZ in differentiated cells to maintain cell quiescence. YAP and TAZ are activated to promote stem/progenitor cell self-renewal and tissue after Hippo pathway suppression. Taking into the consideration that these effectors are implicated in cancer initiation and development, what do we know about Hippo pathway and cellular assays such as apoptosis and proliferation? I think that this section needs to be implemented in light of therapeutic solutions in NSCLC.

Response 5: We had added a paragraph discussing about YAP-induced proliferation and anti-apoptosis in NSCLC in revised manuscript as suggested.

In cellular assays, the Hippo/YAP pathway is physiologically critical for promoting proliferation and regulation of anti-apoptosis in normal cells such as cardiomyocyte and endometrial stromal cells [20,21]. Two previous studies demonstrated important findings that YAP-induced anti-apoptosis worsens the initial treatment response to BRAF and MEK inhibitors and EGFR-TKIs in NSCLC cells with K-ras, BRAF-V600E or EGFR mutations [22,23]. The two studies showed that inhibition of YAP either by genic or pharmacological ablation restores the cytotoxicity of BRAF and MEK inhibitors and EGFR-TKIs to NSCLC cells [22,23]. The findings of the two studies indicated that YAP-induced proliferation and anti-apoptosis are important to the treatment resistance in NSCLC.

This discussion paragraph was added in chapter 3, page 9 line 298-304 in revised manuscript

Reference:

  1. Nguyen, D.X.; Chiang, A.C.; Zhang, X.H.; Kim, J.Y.; Kris, M.G.; Ladanyi, M.; Gerald, W.L.; Massagué, J. WNT/TCF signaling through LEF1 and HOXB9 mediates lung adenocarcinoma metastasis. Cell. 2009, 138, 51–62.

  1. Alamgeer, M.; Peacock, C.D.; Matsui, W.; Ganju, V.; Watkins, D.N. Cancer stem cells in lung cancer: Evidence and controversies. Respirology. 2013 Jul;18(5):757-64.

  1. Bach, D.H.; Luu, T.T.; Kim, D.; An, Y.J.; Park, S.; Park, H.J.; Lee, S.K. BMP4 Upregulation Is Associated with Acquired Drug Resistance and Fatty Acid Metabolism in EGFR-Mutant Non-Small-Cell Lung Cancer Cells. Mol Ther Nucleic Acids. 2018 Sep 7;12:817-828.

  1. Chen, H.; Zhang, M.; Zhang, W.; Li, Y.; Zhu, J.; Zhang, X.; Zhao, L.; Zhu, S.; Chen, B. Downregulation of BarH-like homeobox 2 promotes cell proliferation, migration and aerobic glycolysis through Wnt/β-catenin signaling, and predicts a poor prognosis in non-small cell lung carcinoma. Thorac Cancer. 2018 Mar;9(3):390-399.

  1. Zhang, S.; Wang, Y.; Mao, J.H.; Hsieh, D.; Kim, I.J.; Hu, L.M.; Xu, Z.; Long, H.; Jablons, D.M.; You, L. Inhibition of CK2α down-regulates Hedgehog/Gli signaling leading to a reduction of a stem-like side population in human lung cancer cells. PLoS One. 2012;7(6):e38996.

  1. Zhang, S.; Long, H.; Yang, Y.L.; Wang, Y.; Hsieh, D.; Li, W.; Au, A.; Stoppler, H.J.; Xu, Z.; Jablons, D.M.; et al. Inhibition of CK2α down-regulates Notch1 signalling in lung cancer cells. Version 2. J Cell Mol Med. 2013 Jul;17(7):854-62.

  1. Azzolin, L.; Panciera, T.; Soligo, S.; Enzo, E.; Bicciato, S.; Dupont, S.; Bresolin, S.; Frasson, C.; Basso, G.; Guzzardo, V.; et al. YAP/TAZ incorporation in the β-catenin destruction complex orchestrates the Wnt response. Cell. 2014 Jul 3;158(1):157-70.

  1. Zheng, Y.W.; Li, Z.H.; Lei, L.; Liu, C.C.; Wang, Z.; Fei, L.R.; Yang, M.Q.; Huang, W.J.; Xu, H.T. FAM83A Promotes Lung Cancer Progression by Regulating the Wnt and Hippo Signaling Pathways and Indicates Poor Prognosis. Front Oncol. 2020 Mar 5;10:180.

  1. Kim, W.; Khan, S.K.; Gvozdenovic-Jeremic, J.; Kim, Y.; Dahlman, J.; Kim, H.; Park, O.; Ishitani, T.; Jho, E.H.; Gao, B.; et al. Hippo signaling interactions with Wnt/β-catenin and Notch signaling repress liver tumorigenesis. J Clin Invest. 2017 Jan 3;127(1):137-152.

  1. Isago, H.; Mitani, A.; Mikami, Y.; Horie, M.; Urushiyama, H.; Hamamoto, R.; Terasaki, Y.; Nagase, T. Epithelial Expression of YAP and TAZ Is Sequentially Required in Lung Development. Am J Respir Cell Mol Biol. 2020 Feb;62(2):256-266.

  1. Neto, F.; Klaus-Bergmann, A.; Ong, Y.T.; Alt, S.; Vion, A.C.; Szymborska, A.; Carvalho, J.R.; Hollfinger, I.; Bartels-Klein, E.; Franco, C.A.; et al. YAP and TAZ regulate adherens junction dynamics and endothelial cell distribution during vascular development. Elife. 2018 Feb 5;7:e31037.

  1. Felley-Bosco, E.; Stahel, R. Hippo/YAP pathway for targeted therapy. Transl Lung Cancer Res. 2014 Apr;3(2):75-83.

13 Malik, S.A.; Khan, M.S.; Dar, M.; Hussain, M.U.; Shah, M.A.; Shafi, S.M.; Mudassar, S. Molecular Alterations and Expression Dynamics of LATS1 and LATS2 Genes in Non-Small-Cell Lung Carcinoma. Pathol Oncol Res. 2018 Apr;24(2):207-214.

  1. Raj, N.; Bam, R. Reciprocal Crosstalk Between YAP1/Hippo Pathway and the p53 Family Proteins: Mechanisms and Outcomes in Cancer. Front Cell Dev Biol. 2019 Aug 9;7:159.

  1. Dubois, F.; Keller, M.; Calvayrac, O.; Soncin, F.; Hoa, L.; Hergovich, A.; Parrini, M.C.; Mazières, J.; Vaisse-Lesteven, M.; Camonis, J.; et al. RASSF1A Suppresses the Invasion and Metastatic Potential of Human Non-small-cell Lung Cancer Cells by Inhibiting YAP Activation through the GEF-H1/RhoB Pathway. Cancer Res. 2016, 76, 1627–1640.

  1. Huang, H.; Zhang, W.; Pan, Y.; Gao, Y.; Deng, L.; Li, F.; Li, F.; Ma, X.; Hou, S.; Xu, J.; et al. YAP Suppresses Lung Squamous Cell Carcinoma Progression via Deregulation of the DNp63-GPX2 Axis and ROS Accumulation. Cancer Res. 2017 Nov 1;77(21):5769-5781.

  1. Wu, T.; Hu, H.; Zhang, T.; Jiang, L.; Li, X.; Liu, S.; Zheng, C.; Yan, G.; Chen, W.; Ning, Y.; et al. miR-25 Promotes Cell Proliferation, Migration, and Invasion of Non-Small-Cell Lung Cancer by Targeting the LATS2/YAP Signaling Pathway. Oxid Med Cell Longev. 2019 Jun 18;2019:9719723.

  1. Luo, H.M.; Wu, X.; Xian, X.; Wang, L.Y.; Zhu, L.Y.; Sun, H.Y.; Yang, L.; Liu, W.X. Calcitonin gene-related peptide inhibits angiotensin II-induced NADPH oxidase-dependent ROS via the Src/STAT3 signalling pathway. J Cell Mol Med. 2020 May 5.

  1. Chung, L.Y.; Tang, S.J.; Wu, Y.C.; Yang, K.C.; Huang, H.J.; Sun, G.H.; Sun, K.H. Platinum-based combination chemotherapy triggers cancer cell death through induction of BNIP3 and ROS, but not autophagy. J Cell Mol Med. 2020 Jan;24(2):1993-2003.

  1. Song, Y.; Fu, J.; Zhou, M.; Xiao, L.; Feng, X.; Chen, H.; Huang, W. Activated Hippo/Yes-Associated Protein Pathway Promotes Cell Proliferation and Anti-apoptosis in Endometrial Stromal Cells of Endometriosis. J Clin Endocrinol Metab. 2016 Apr;101(4):1552-61.
  2. Lin, Z.; Zhou, P.; von Gise, A.; Gu, F.; Ma, Q.; Chen, J.; Guo, H.; van Gorp, P.R.; Wang, D.Z.; Pu, W.T. Pi3kcb links Hippo-YAP and PI3K-AKT signaling pathways to promote cardiomyocyte proliferation and survival. Circ Res. 2015 Jan 2;116(1):35-45.
  3. Lin, L.; Sabnis, A.J.; Chan, E.; Olivas, V.; Cade, L.; Pazarentzos, E.; Asthana, S.; Neel, D.; Yan, J.J.; Lu, X.; et al. The Hippo effector YAP promotes resistance to RAF- and MEK-targeted cancer therapies. Nat. Genet. 2015, 47, 250–256.
  4. Kurppa, K.J.; Liu, Y.; To, C.; Zhang, T.; Fan, M.; Vajdi, A.; Knelson, E.H.; Xie, Y.; Lim, K.; Cejas, P.; et al. Treatment-Induced Tumor Dormancy through YAP-Mediated Transcriptional Reprogramming of the Apoptotic Pathway. Cancer Cell. 2020 Jan 13;37(1):104-122.e12.

Reviewer 4 Report

Ping-Chih Hsu and colleagues reviewed the involvement of src signaling and Hippo/Yap signaling in NSCLC. Furthermore authors collected evidences of a crosstalk between Src and Hippo/YAP pathways. Lastly, Ping-Chih Hsu and colleagues suggested new therapeutic strategies to overcome targeted therapy resistance and improve survival of NSCLC patients.

I suggested authors to include some tables to better summarize clinical and preclinical evidences of Src and/or Hippo/Yap pathways involvement. Tables should include some specifications as e.g. patient alterations (EGFR mutations and which, ALK mutations, any detected mutation), type of treatment, primary resistance or not and obviously references. Authors could include a table for each Chapter (chapter 2, 3, 4). Authors could also fill a table about preclinical evidences as Supplementary material.

Of note, from a clinical point of view authors could also evidence if src and hippo/Yap pathways involvement is more frequent in presence of specific EGFR mutations: e.g common or uncommon and which one. This could offer good suggestions to clinicians to start new clinical trials.

A recent paper published on Cancer Research (Slemmons et al.) suggested the use of DNA methyltransferase inhibitor and dasatinib against rhabdomyosarcoma. Authors could insert a paragraph or a comment in Chapter 5 about the advantage of DNA methyltransferase inhibitors in src and Hippo/Yap crosstalking.

Lastly, I suggested to delete the paragraph 5.2 since authors previously described the relevance of SRC and Hippo/YAP crosstalk as consequence/cause of EGFR-TKIs resistance. Thus, I think that the use of this TKI can not be considered a potential therapy targeting Src-YAP.

Author Response

Response to Reviewer 4 Comments

Ping-Chih Hsu and colleagues reviewed the involvement of src signaling and Hippo/Yap signaling in NSCLC. Furthermore authors collected evidences of a crosstalk between Src and Hippo/YAP pathways. Lastly, Ping-Chih Hsu and colleagues suggested new therapeutic strategies to overcome targeted therapy resistance and improve survival of NSCLC patients.:

Point 1: I suggested authors to include some tables to better summarize clinical and preclinical evidences of Src and/or Hippo/Yap pathways involvement. Tables should include some specifications as e.g. patient alterations (EGFR mutations and which, ALK mutations, any detected mutation), type of treatment, primary resistance or not and obviously references. Authors could include a table for each Chapter (chapter 2, 3, 4). Authors could also fill a table about preclinical evidences as Supplementary material.

Response 1: Done as suggested

We added table 1 and table 2 to summarize the Src and Hippo/YAP related targeted therapy resistance.

Chemical compounds and drugs targeting Src-YAP axis is summarized in table 3.

Point 2:  Of note, from a clinical point of view authors could also evidence if src and hippo/Yap pathways involvement is more frequent in presence of specific EGFR mutations: e.g common or uncommon and which one. This could offer good suggestions to clinicians to start new clinical trials.

Response 2: In response to the point about if src and hippo/Yap pathways involvement is more frequent in presence of specific EGFR mutations, Src-YAP signaling appears as primary or acquired resistances to EGFR-TKIs in NSCLC patients with common mutations (L858R and exon 19 deletion).

In the previous study conducted by Fan et al., acquired YES1 amplification was detected in 5 EGFR mutant NSCLC patients (3 had L858R mutation and 2 had exon 19 deletion mutation) who were pre-treated with EGFR-TKIs erlotinib or afatinib and had acquired resistance to EGFR-TKIs [1]. Another study by Chaib et al. found that EGFR mutant lung cancer cells survive initial EGFR-TKI therapy through the co-activation of STAT3 and Src–YAP signaling. In a cohort analysis of 64 EGFR mutant NSCLC patients (62 had common mutations L858R and exon 19 deletion, the other 2 had uncommon mutations L861Q and G719X) treated with first-line EGFR-TKIs, patients with a high expression of STAT3 or YAP1 had a worse progression-free survival in EGFR-TKI therapy. Treatment with the EGFR-TKI gefitinib, in combination with the SRC inhibitor saracatinib (AZD05300), was shown to have a synergistic antitumor effect on NSCLC cells through in vitro and in vivo experiments presented in the same study [2]. Taken together, Src–YAP signaling appears as primary and acquired resistances to EGFR-TKIs in NSCLC patients with sensitive EGFR mutations (L858R and exon 19 deletions). EGFR-TKIs in combination with a therapy targeting Src-YAP axis may overcome the resistance to EGFR-TKIs, and future clinical trials investigating this combination in EGFR-mutant NSCLC patients are warranted.

This paragraph was added in chapter 4, page 12 line 400-418 in revised manuscript

Point 3: A recent paper published on Cancer Research (Slemmons et al.) suggested the use of DNA methyltransferase inhibitor and dasatinib against rhabdomyosarcoma. Authors could insert a paragraph or a comment in Chapter 5 about the advantage of DNA methyltransferase inhibitors in src and Hippo/Yap crosstalking.

Response 3: We added a paragraph to discuss the combination of DNA methyltransferase inhibitor and dasatinib, and cited the refence in revised manuscript as suggested [3].

The paragraph was added in chapter 5.1 page 13 line 451-454 and page 14 line 455-457

Point 4: Lastly, I suggested to delete the paragraph 5.2 since authors previously described the relevance of SRC and Hippo/YAP crosstalk as consequence/cause of EGFR-TKIs resistance. Thus, I think that the use of this TKI can not be considered a potential therapy targeting Src-YAP.

Response 4:

We deleted the paragraph 5.2 as suggested in revised manuscript.

Reference:

  1. Fan, P.D.; Narzisi, G.; Jayaprakash, A.D.; Venturini, E.; Robine, N.; Smibert, P.; Germer, S.; Yu, H.A.; Jordan, E.J.; Paik, P.K.; et al. YES1 amplification is a mechanism of acquired resistance to EGFR inhibitors identified by transposon mutagenesis and clinical genomics. Proc Natl Acad Sci U S A. 2018 Jun 26;115(26):E6030-E6038.

  1. Chaib, I.; Karachaliou, N.; Pilotto, S.; Codony Servat, J.; Cai, X.; Li, X.; Drozdowskyj, A.; Servat, C.C.; Yang, J.; Hu, C.; et al. Co-activation of STAT3 and YES-Associated Protein 1 (YAP1) Pathway in EGFR-Mutant NSCLC. J Natl Cancer Inst. 2017 Sep 1;109(9): djx014.

  1. Slemmons, K.K.; Yeung, C.; Baumgart, J.T.; Martinez Juarez, J.O.; McCalla, A.; Helman, L.J. Targeting Hippo-dependent and Hippo-independent YAP1 signaling for the treatment of childhood rhabdomyosarcoma. Cancer Res. 2020 Apr 30:canres.3853.2019.

Round 2

Reviewer 4 Report

Accepted